# Gut dysbiosis narrative in psoriasis: matched-pair approach identifies only subtle shifts correlated with elevated fecal calprotectin

Bayazit Yunusbayev,[1,2] Anna Bogdanova,[3] Nadezhda Nadyrchenko,[4] Lavrentii Danilov,[2] Viktor Bogdanov,[5,6] Grigory Sergeev,[3] Radick Altinbaev,[7] Fanil Bilalov,[8,9] Milyausha Yunusbaeva[1,2]

**ABSTRACT**   Many studies have reported gut microbiome alterations in psoriasis patients, suggesting dysbiosis. While evidence for dysbiosis and its link to pathogenesis remains inconclusive, murine models of psoriasis suggest that gut microbiome alterations develop in response to psoriasis-like inflammation. Hence, the dominant narrative about gut microbiome alterations' impact on disease should be evaluated critically with more data and a well-powered approach. In this case-control study, we used deep sequencing of fecal samples from 53 psoriasis patients and 47 healthy donors to reconstruct the strain/species-level content of the gut microbiome. Unlike previous studies, we first identified matched pairs for each patient with healthy donors to adjust for microbiome variability and increase power. We found no evidence for depleted gut community diversity and apparent divergence in structure between patients and healthy individuals. However, our matched-pair approach identified a subtle but systematic increase in select bacteria among patients, e.g., *Megasphaera elsdenii* and *Eubacterium CAG 180*. We next showed that these enriched species were correlated with elevated biomarkers of intestinal and systemic inflammation and liver function. Functionally, one of the top species, *Megasphaera elsdenii*, is a potent lactate utilizer in the context of intestinal lactic acidosis and inflammation. While our findings hardly support overt dysbiosis in the large intestine, the observed microbial changes correlate with moderately elevated calprotectin, albeit at levels not enough to diagnose ongoing inflammation. Hence, the sources of elevated inflammatory markers in patients' intestines remain unclear and warrant further investigation to clarify their cause-and-effect relationship with the disease.

**IMPORTANCE**   With sufficient taxonomic resolution and sample size, this study critically evaluates new and published data on the gut microbiome in psoriasis patients. It shows that observed taxonomic changes in patients are modest and do not meet strict criteria for gut dysbiosis, at least in the large intestine. Instead, observed taxonomic changes in psoriasis patients can be explained by the microbial response to possible low-grade inflammation with unknown localization in the intestine and unclear impact on the host. The authors point out that published endoscopic data point to the small intestine as the site of gut inflammation. Therefore, further research focused on the small intestine would be informative to clarify the hypothetical gut-psoriasis link.

**KEYWORDS**   gut microbiota, dysbiosis, low-grade Inflammation, psoriasis, lactate

**Peer Reviewer** Jian Zhou, Chinese Academy of Sciences, Beijing, China

Address correspondence to Bayazit Yunusbayev, yunusbb@gmail.com.

Bayazit Yunusbayev, Anna Bogdanova, and Milyausha Yunusbaeva contributed equally to this article. Author order was determined on the basis of seniority.

The authors declare no conflict of interest.

See the funding table on p. 13.

Growing evidence suggests that human autoimmune conditions are accompanied by disturbed gut microbiota, but the causal relationship between many diseases and disturbed gut immunity is poorly understood (1, 2). Our mechanistic understanding

of the host microbiome interplay is derived from experiments on model organisms, rather than from human subjects with autoimmune conditions. For instance, studies on model organisms show that gut commensals modulate gut mucosal immunity with consequences at a systemic level. Specifically, systemic effects dysregulate the T helper 17 immune pathway in distal tissues (3), an immune effector pathway contributing to psoriasis immunopathology (4). While this fundamental insight from model organisms is intriguing, we still need to accumulate and evaluate evidence from observational case-control studies for autoimmune diseases and psoriasis patients.

For psoriasis, dozens of case-control studies reported altered gut microbiomes, but data so far are inconsistent about possible dysbiosis in the large intestine and enrichment of specific taxa (5). We summarized previous studies to evaluate two major questions in case-control microbiome studies—evidence for gut dysbiosis and evidence for bacterial species enriched in patients (Table S1). Our review allowed us to highlight several issues. Gut microbiome studies in psoriasis suggest dysbiosis but often equate any community alterations with dysbiosis without presenting data on the signs of dysbiosis. Namely, signs of dysbiosis such as a bloom of pathobionts, loss of commensals (or keystone species), or loss of diversity (6, 7) that would meet the classical definitions tested by time (8). For instance, when we asked whether microbiome studies for psoriasis met at least one dysbiosis criteria, i.e., diversity loss in patients, only work by Hidalgo-Cantabrana et al. (5) reported alpha diversity decline in patients (Table S1). Most other studies (8 out of 10 that reported alpha diversity) found no evidence of a decline in microbial diversity in patients compared to healthy controls (Table S1). Similarly, when we tried to identify bacterial taxa reproducibly associated with patients, only *Megasphaera* species featured in more than one study. *Megasphaera* was featured in our study and reference (9), both based on shotgun sequencing.

On the other hand, discussing sources of observed discrepancies is premature before we accumulate uniformly processed case-control data sets from identical populations. Indeed, most published studies are underpowered, report patients from different continental populations (China, South and North America, Central Asia, and Europe), and characterize microbiomes at different taxonomic resolutions. Therefore, further studies with larger samples from one regional population and uniform study protocols (sequencing depth and statistical analyses) would be needed to yield consistent, reproducible evidence for dysbiosis in the large intestine, if any. More importantly, it is still unclear whether gut microbiome alterations in psoriasis are accompanied by inflammation in the large and small intestines. If so, whether such alterations causally impact pathogenesis or result from an inflammatory process in psoriasis. This study used deep shotgun sequencing to examine fecal-derived gut microbiota in 53 psoriasis patients and healthy controls. We evaluated the evidence for dysbiosis and compared microbial composition with intestinal and systemic inflammation markers.

## MATERIALS AND METHODS

### Samples and quality control

In this case-control study, we collected freshly frozen fecal samples from 47 healthy adult donors and 53 psoriatic patients admitted to the Republic Dermatovenerologic Dispensary in 2018 (Ufa, Russia). Metagenomic sequences for the 47 healthy donors were released earlier in our other study.(10) With this total sample size of 100 donors, our study objective was to test if patients had systematically abundant gut bacteria. We first computed group-wise effect size for psoriasis on gut microbiome using the evident tool (11). Our estimates suggest that psoriasis has a small effect size (Cohen's $d = 0.37$) on the gut microbiome. As a rule of thumb, Cohen's $d$ values of 0.2, 0.5, and 0.8 are generally considered "small," "medium," and "large" effect sizes, respectively. With this effect size estimate, we performed power analysis at a significance level of 0.05 for a range of sample sizes from 10 to 400 (Fig. S1). It is clear that to detect group-wise differences between cases and controls, one needs over 200 donors to achieve a reasonable

power of 0.8 (Fig. S1), while 100 samples would be underpowered. In theory, one can increase statistical power by matching cases to controls. We, therefore, implemented a matched-pair approach (see below) to identify differentially abundant species using case-control pairs with similar microbiome profiles. To minimize factors that strongly affect microbiota composition, we interviewed patients and healthy donors before enrolling in the study cohort. Table 1 shows the demographics of all of the participants. Our detailed questionnaire was designed to exclude prior antibiotic and contraceptive usage, alcohol intake, and specific diets. Healthy volunteers who reported allergic or inflammatory conditions other than psoriasis were also excluded due to evidence for association with gut dysbiosis. Each donor's fecal-derived bulk DNA was extracted using the QIAamp DNA Stool Mini Kit (Qiagen, the Netherlands). Fecal samples were also examined for calprotectin levels. Measurement of fecal calprotectin was performed using the CALPROLAB Calprotectin ELISA (ALP) Kit (Svar Life Science, Norway) according to the manufacturer's protocol (CalproLab ELISA, #CALP0170). Final ELISA readouts were expressed as micrograms of calprotectin per gram of fecal material. Deep metagenomic sequencing (30–50 million sequences per donor) was carried out using Illumina 150 bp paired-end sequencing on the Novaseq6000 platform.

## Raw sequence quality control and data preprocessing

We examined sequence quality before and after quality trimming and decontamination using FastQC version 0.11.9 (12). Trimmomatic version 0.33 was used to remove low-quality bases (Q20) and clip Illumina adapters from the raw sequences (13). For Trimmomatic, the following parameters were used: ILLUMINACLIP:NexteraPE-PE.fa:2:30:10:8:TRUE SLIDINGWINDOW:4:20 MINLEN:74. We additionally removed tandem repeats using Tandem Repeats Finder version 4.09 (14). Finally, the KneadData version 0.10.0 pipeline of the bioBakery toolset was used to decontaminate reads originating from the human genome, transcriptome, and microbial RNA (15). Specifically, we used the human reference genome (build hg37), the human reference transcriptome (build hg38), and the SILVA ribosomal RNA reference databases.

## Bacterial, archaeal, and fungal taxonomic composition

The taxonomic composition of the fecal-derived microbial community was predicted at the species level based on the MetaPhlAn version 3.0 tool using default parameters (16). Bacterial, fungal, and archaeal taxons observed only in three donors and at a relative abundance of less than 0.0001 were discarded before all downstream analyses except for diversity estimation. This threshold ensures that we keep rare bacterial species that may impact autoimmune conditions, as demonstrated in a recent work (17).

## Bacterial community diversity and principal component analysis

Shannon alpha and beta diversity measures were estimated using the microbiome R package (18). Differences in Shannon alpha diversity between patients and controls were tested using the two-sample Kolmogorov-Smirnov test. The Kolmogorov-Smirnov test was performed using the two-sided hypothesis in the ks.test() function of the stats R package (19). To visualize gut community differences between donors using major axes of microbial variation, we applied principal component analysis (PCA) on the species abundance matrix. To adapt sparse data on relative abundance for principal component

**TABLE 1** Baseline demographics of the study cohort

| Characteristics | Healthy | Psoriasis | *P*-value |
| --- | --- | --- | --- |
| Sample size | 47 | 53 | NA[a] |
| Sex (% female) | 51 | 53 | NA |
| Age (mean age) (years) | 33.41 | 38.89 | 0.00178 |
| Mean BMI (kg/m$^2$) | 22.8 | 26.5 | 0.00005 |

[a] NA - Not assessed.

analysis, we imputed zeros in the relative abundance table using the cmultRepl() function in the zCompositions R package version 1.4.0-1 (20). Next, to take composition-ality into account, the relative abundance table was transformed using the center log ratio (CLR) before PCA. PCA was carried out using the prcomp() function in the stats R package (19).

## Testing group-level taxonomic differences

To identify differentially abundant species, we used both conservative (such as ALDEx2) and less conservative statistical procedures (such as LefSe) suitable for exploratory purposes. First, we used a more stringent ANOVA-like differential expression method (R package ALDEx2, version 1.30.0) with default parameters. This method allows for comparing case-control differences to intra-group dispersion. Next, we inferred taxonomic differences between patients and controls using popular linear discriminatory analysis with effect size estimation (LefSe) implemented in the microbial R package.

## Constructing matched pairs with similar taxonomic profiles

Gut microbial composition varies extensively from donor to donor. Extensive variabil-ity can obscure disease-associated taxonomic shifts. To level out the effect of this interindividual variability and increase statistical power, we matched each psoriasis patient with a suitable healthy donor with a similar core microbiome. It is essential to clarify that when we seek a "similar microbiome," we never find exactly matching microbiomes. We aim to find a pair of patients and controls as closely as possible to focus on the remaining differences that might be unique to patients. This approach is akin to matched-pair design, and we outline its principle using a schematic diagram (Fig. S5). Thus, we estimated Euclidean distances between all the donors in the data set to identify the so-called matched case-control pairs with similar core microbiomes. Euclidean distances were computed by summarizing 11 principal components cumula-tively, capturing 70% of the microbial variability. By projecting each patient on Euclidean space, we identified its closest healthy neighbor using the nearest neighbor algorithm. Euclidean distances and matched pairs were computed and identified using the MatchIt version 4.5.4 R package (21). We identified 47 case-control pairs, which were used to test for systematic taxonomic shifts across matched pairs. Systematic shifts were tested using the Wilcoxon rank sum test for matched pairs using the microbiotaPair wrapper R function (22). The microbiotaPair R wrapper function performs the Wilcoxon test using the standard R wilcox.test(). It reports a two-sided alternative hypothesis *P*-value and a false discovery rate (FDR). The matched-pair approach implemented in our study can be reproduced using the code and notebook ("3_matching_pairs_anal-ysis.ipynb") available at https://github.com/Annanielle/Gut-microbiota-in-Psoriasis and https://github.com/yunusbb/Gut-microbiota-in-Psoriasis.

## Testing association between top 14 differentially abundant species and host biomarkers

For each patient and healthy donor, we measured 35 host biomarkers to assess evidence for intestinal inflammation (fecal calprotectin), liver function (alanine transaminase, ALT, and aspartate transferase, AST), and other blood biomarkers, such as c-reactive protein (the complete list is given in Fig. S2 and S3). The association between host biomarkers' levels, bacterial species abundance, and diagnosis was assessed using general linear models in the MaAsLin2 tool version 1.12.0 with default parameters (23). MaAsLin2 reports *P*-values and estimates of *q*-values, i.e., false discovery rate. Associations were deemed significant at 10% FDR. We also explored the distribution of each host biomarker in patients and controls using box and whisker plots. Between-group differences in biomarker levels were tested using the Wilcoxon test (function wilcox.test() in R) (Fig. S2 and S3). Next, partial least squares discriminant analysis (sPLS-DA) was used to evaluate how well host biomarkers can discriminate patients and controls. The sPLS-DA

was performed using the mixOmics R package (24). Finally, we estimated the Kendall correlation index between the top psoriasis-associated species and host biomarkers using the cor() method from the base R package.

## RESULTS

### Microbiome composition based on group-wise comparisons

In this case-control study, we recruited 47 healthy adult donors and 53 psoriatic patients, balanced by sex (Table 1). Despite efforts to match by age, patients' average age (mean age = 38.89) was slightly higher than in healthy donors (mean age = 33.41). Patients also had somewhat higher body mass index (BMI) (mean BMI = 26.5) compared to healthy donors (mean BMI = 22.8) (Table 1). We projected donors on principal components of microbiome variation to examine whether observed differences in mean values reflect strong enrichments and correlations with the microbiome. Donor distribution on PCA allowed us to explore potential clustering with age, BMI, and sex (Fig. S4). Neither age nor BMI tended to correlate with major principal components (axes) of microbiome variation (Fig. S4).

By focusing on Bacteria, Archaea, and Eukaryotes/Fungi, we identified 424 microbial taxa in our combined set of 53 patients and 47 controls using the MetaPhlAn 3.0 taxonomic inference tool (16). Consistent with most earlier studies (Table S1), patients had no apparent reduction in taxonomic diversity (Fig. 1A and B) when we analyzed unfiltered diversity. We filtered rare taxa encountered in only three individuals to reduce noise and analyzed 249 microbial taxa in all downstream analyses. We started by asking whether the most variable taxa in the data set separate patients from healthy donors using PCA. The top two principal components (PCs) that capture the most variable microbial species did not separate patients from controls and explained only a minor portion (13.4%) of the total variation (Fig. 1C). Most of the microbial variation is captured by other PCs that describe species' relative abundance among smaller subsets of individuals, perhaps down to inter-individual variation. Our PC analysis, therefore, emphasizes the common challenge with gut microbiome derived from fecal material—extensive inter-individual variability (25). Next, we used the ALDEx2 method to explore how within-group variability (dispersion within cases and patients) compares to case-control differences. When species (gray dots in Fig. 1D) are stratified based on

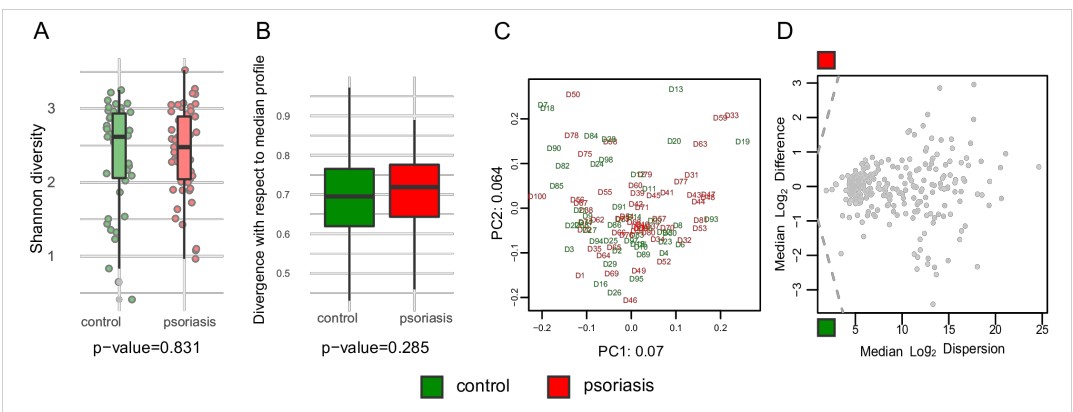

**FIG 1** Gut microbiome diversity and composition in patients ($n = 53$) and healthy controls ($n = 47$). (A) Alpha diversity (Shannon index) distribution among 53 patients and 47 healthy donors. The box represents the interquartile range, the horizontal line inside shows the median, the whiskers indicate lower to upper quartiles, and points outside the whiskers indicate potential outliers. No significant differences in diversity between the patients and control groups ($P = 0.83$, Kolmogorov-Smirnov two-sided hypothesis test). (B) Beta diversity (Bray-Curtis dissimilarity) distribution. The box-and-whiskers plot has the same elements as in panel A. No significant divergence between groups ($P = 0.285$). (C) PCA based on bacterial species proportions. Proportions were CLR transformed. PC1 and PC2 are given with the proportion of total variance explained. (D) ALDEx analysis. Median between-group differences stratified by within-group dispersion. The dashed diagonal line indicates points where within-group differences equal between-group differentiation. None of the between-group differences exceeded within-group dispersion, i.e., no group-level significant differences.

their dispersion (*x*-axis), case-control median differences (*y*-axis) never exceed dispersion: no dots are observed on the upper left or bottom left corner outside the dashed line. Our ALDEx2 analysis suggests that extensive intra-group dispersion would obscure any moderate shifts in abundance between patients and controls, even if they have biological effects (Fig. 1D).

## Constructing matched pairs to level out interindividual variability

We sought to identify pairs of patients and controls with similar microbial profiles to account for inter-individual variability in microbial composition. Similar microbial profiles were established based on Euclidean distances between donors. These Euclidean distances were computed from 11 principal components, cumulatively capturing much of the microbial variation. Using the nearest-neighbor matching method, we constructed 47 case-control pairs with matched microbial profiles (see schematic diagram of the matched-pair approach in Fig. S5 and Fig. 2A and details in Materials and Methods). Several microbiome studies have shown that matching patients with controls increases statistical power in detecting abundance shifts associated with disease (26–28). Our matched-pair analysis highlighted 14 bacterial species with abundance shifts in patients (Wilcoxon signed-rank test, $P \leq 0.05$). Namely, seven species had systematically higher proportions in patients, and the other seven in matched controls (Fig. 2; Table S2). These species would have been missed if we had relied on ALDEx2, a stringent test that seeks group-wise differences that surpass intra-group variability.

We allowed for some false discoveries (see FDR for each species [Table S2]) to consider more candidate species for downstream comparisons. We also performed a popular linear discriminant analysis, LEfSe, to allow comparisons with earlier studies. Four species (*Megasphaera elsdenii*, *Eubacterium* sp. *CAG 180*, *Bacteroides xylanisolvens*, and *Prevotella* sp. *AM42 24*) identified by our matched pair approach were also identifiable as patient-associated by LEfSe (Fig. S6). Finally, we asked if our seven species-associated patients were reported earlier in published data by considering taxonomic resolution (Table S1). None of the studies based on 16S sequencing showed overlap with our seven-patient associated species, presumably due to low taxonomic resolution. Among the other five studies (out of 12 summarized) that reported species-level data using shotgun sequencing, only Chang et al. (29) and Xiao et al. (9, 29) reported some overlaps. Namely, Chang et al. (29) and Xiao et al. (9, 29) reported an increased proportion of *Megasphaera unclassified* among patients and association of *Bacteroides xylanisolvens* patient-dominated cluster (PSO2) (9, 29). Thus, contrary to the earliest studies (30), our and most published data (summarized in Table S1) imply no core disease-specific

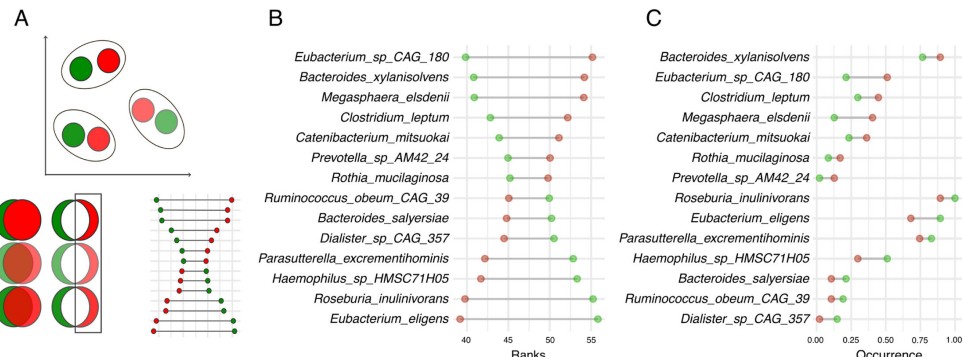

**FIG 2** Differentially abundant bacteria in case-control matched pairs. (A) Schematic representation of matched-pair construction procedure and Wilcoxon test for matched pairs. Patients (red circles) and healthy controls (green circles) are paired based on similarity of microbial profiles, i.e., closeness on the Euclidean space based on 11 principal components of microbial abundance. Next, matched pairs are tested to find differentially abundant species using the Wilcoxon test: (B) differentially abundant bacteria in matched pairs (*N* = 47 matched pairs) ordered by ranks (*P* ≤ 0.05); (C) differentially abundant bacteria ordered by occurrence in matched pairs (*N* = 47 matched pairs) (*P* ≤ 0.05).

community. Instead, different bacterial taxa features in various studies are consistent with the extensive variability of gut microbiome and functional redundancy (31).

## Association between candidate species and host biomarkers of intestinal inflammation

In our study cohort, fecal calprotectin was, on average, elevated in patients ($P$ = 0.008, Fig. 3A). The higher average estimate in the patient group was explained by a higher proportion of patients (15/53) than controls (1/47) with calprotectin over 35 mcg/g. In this study, we operationally defined an elevated calprotectin level if it was higher than 35 mcg/g. This calprotectin threshold is close to the lower bound (<40 mcg/g), which is used to exclude inflammatory bowel disease. On the other hand, this threshold intersects the upper range of normal variation (~10–50 mcg/g) among healthy individuals (32). Since fecal calprotectin indicates an altered environment in the patient's intestines, we tested if our 14 species are correlated with the readout for intestinal inflammation. We used linear models in the MaAsLin tool (23). We found that four out of seven species (*Megasphaera elsdenii*, *Catenibacterium mitsuokai*, *Eubacterium* sp. *CAG:180*, and *Rothia mucilaginosa*) that are elevated in patients were positively associated with the elevated levels of fecal calprotectin (red boxes with plus sign, Fig. 3B). Conversely, one of the seven species (*Eubacterium eligens*) characteristic for healthy controls was associated with reduced levels (values less than 35 mcg/g) of fecal calprotectin (blue box with a negative sign, Fig. 3B). We next ranked patients and controls based on calprotectin levels and graphed proportions of the seven species associated with psoriasis (Fig. 3C). As expected, the four calprotectin-associated species were elevated in a subset of patients with elevated calprotectin (calprotectin > 35). Notably, one healthy donor with elevated calprotectin (calprotectin = 59.15) showed unusually high proportions of the same calprotectin-associated species (Fig. 3C, stacked bar indicated by asterisk). This healthy donor also had higher blood biomarkers of inflammation (data not shown). Based on these findings, one could speculate that patients with elevated calprotectin (Fig. 3C) could have driven the discovery of four species (out of seven in Fig. 3B) initially identified as elevated in patients in our matched-pair analysis (Fig. 2). What if we started by identifying top bacterial species that follow a linear increase in calprotectin irrespective of diagnosis? For example, we applied redundancy analysis (RDA) to rank bacterial species based on how much they contributed to the calprotectin-associated variation. We then highlighted the top five based on the RDA score (Table S3). These top species included three previously detected taxa based on matched-pair analysis (*Eubacterium* sp. *CAG:180, M. elsdenii,* and *C. mitsuokai*) and two additional species—*Phascolarctobacterium succinatutens* and *Prevotella copri*. Thus, the bacterial species we initially highlighted as diagnosis-associated can also be explained by fecal calprotectin levels, which are more frequent in patients. Finally, we asked if the bacterial species we highlighted as diagnosis-associated correlate with blood markers.

## Association between 14 candidate species and host biomarkers

We first pre-selected blood biomarkers that showed differences ($P \leq 0.05$) between patients and controls when taken individually (Fig. S2). Next, we used the sPLS-DA method to evaluate how useful these biomarkers are in discriminating psoriasis from controls when taken jointly (Fig. 4A). When used jointly, multiple blood biomarkers perform better for chronic inflammatory states. We included calprotectin for comparison (33). We found that best-performing markers (biomarkers highly loading first component) were related to liver function (ALT, AST, cholesterol, and LDL, i.e., low-density lipoprotein) and inflammation in the body (erythrocyte sedimentation rate) and inflammation in the intestine (fecal calprotectin) (Fig. 4A). Finally, we tested whether these blood markers were correlated with the 14 differentially abundant bacteria in patients (Fig. 4B). One can use fecal calprotectin correlation values as a reference and see that blood biomarkers have only a weak tendency to correlate with the same three calprotectin-associated species (indicated in bold, *Eubacterium* sp. *CAG:180, M. elsdenii*,

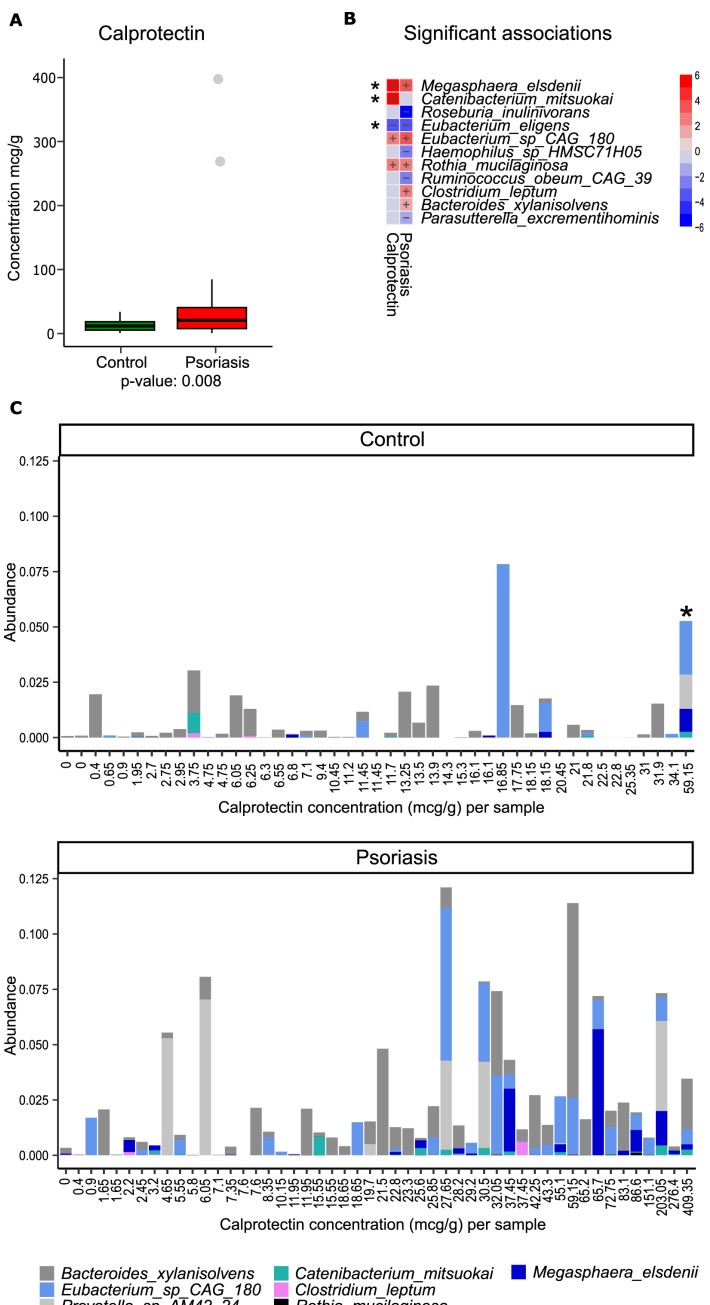

**FIG 3** Fecal calprotectin levels in the study cohort and associated bacterial taxa. (A) Fecal calprotectin distribution among patients and healthy donors. (B) Bacterial taxa associated with fecal calprotectin levels and diagnosis. Association tests were deemed significant at a 10% FDR. Red and blue shadings indicate the extent of positive and negative association, and the plus sign shows statistically significant findings. (C) Relative abundance (fractions of 1) of bacterial taxa associated with diagnosis and ordered by increasing levels of fecal calprotectin in cases and controls.

and *C. mitsuokai*). We conclude that a commonly used panel of blood biomarkers had little use in tracking possible low-grade inflammation in the intestine (elevated fecal calprotectin) and gut microbial shifts. This is true for our cohort of patients with little evidence of large intestinal dysbiosis.

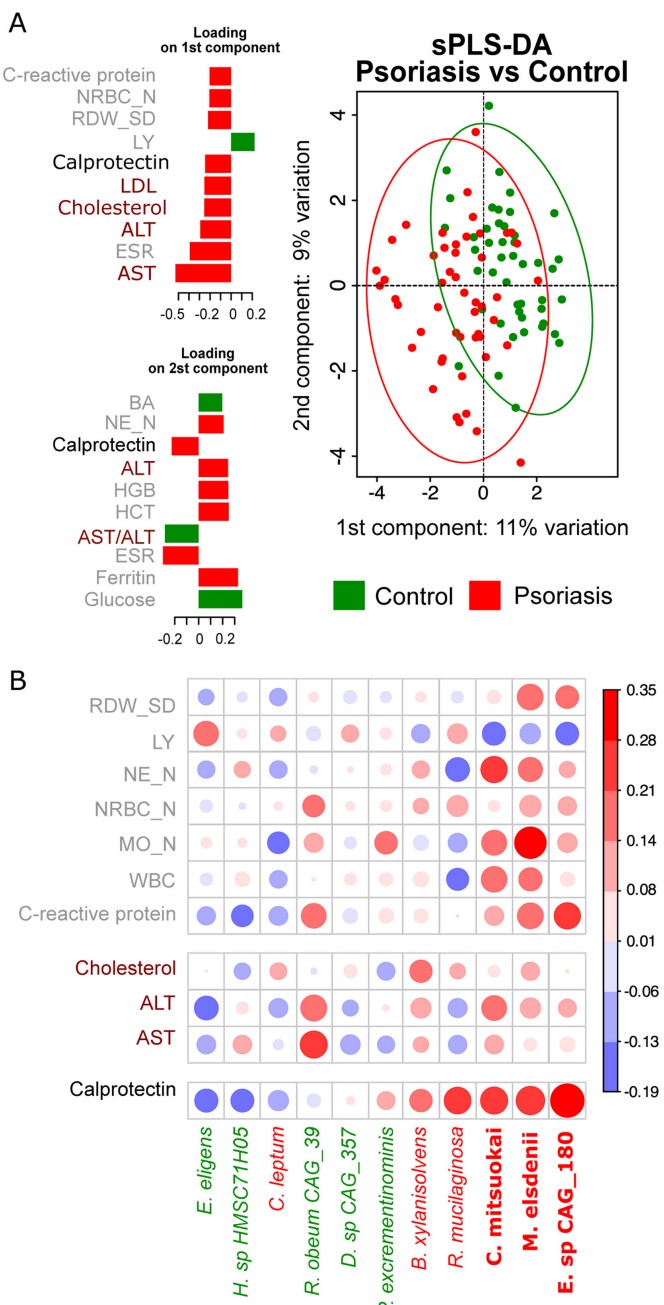

**FIG 4** Host biomarkers and their association with the top 14 bacterial taxa. (A) Least squares discriminant analysis of patients and controls based on the host biomarkers (blood biomarkers and fecal calprotectin). Informative host biomarkers are summarized in two components. Informative biomarkers are ordered based on their absolute loadings on the first and second components. Calprotectin is black; liver function-related biomarkers are brown; other blood biomarkers are gray. (B) Host biomarkers that correlate with the top psoriasis-associated species (see Fig. 2B). Positive (red) and negative (blue) Kendall's tau correlation coefficients are shown with circle size proportional to coefficient value. Bacterial taxa that were previously found to be associated (matched-pair Wilcoxon test, see Fig. 2B) with psoriasis are shown in red and green when associated with healthy controls. Bacterial taxa previously found to be significantly associated with calprotectin (see Fig. 3B) are indicated in bold red.

## DISCUSSION

Current knowledge about the human microbiome in psoriasis and other inflammatory conditions is primarily based on fecal-derived material. Hence, observed taxonomic alterations likely reflect changes in the large intestinal community. This study shows that gut microbiome analysis based on fecal-derived material from patients shows no evidence of depleted microbial diversity compared to healthy control. Among earlier studies, 8 out of 12 case-control comparisons (two did not report Shannon diversity) reported no changes in microbial diversity in patients (Table S1). Thus, one study reported a decline in diversity among patients (5), while another reported an increase in diversity among patients (30). Taken together, more than half of the published studies we reviewed hardly support strong dysbiosis in the large intestine if we use this single criterion (depleted microbial community diversity). Yet, some of these studies summarized in Table S1 tended to interpret observed taxonomic alterations as a signature of dysbiosis without sufficient evidence. In the meantime, to conclude dysbiosis, one has to meet several criteria, such as depleted gut community, strong taxonomic shifts explained by (i) increased proportions of opportunistic gut flora, and (ii) depleted proportions of healthy commensals (6, 7). Our findings show that patients do not have strong taxonomic shifts but tend to have a similar large intestinal community as healthy donors (PCA plot in Fig. 1C), albeit with minor differences (Fig. 2). Again, our findings agree with most earlier studies (9 out of 12) demonstrating similarity between patients and healthy controls. Most earlier studies showed that overall gut community structure overlaps on PCA and PCoA plots (Table S1). Only two studies (out of 12) reported differences between patients and controls strong enough to be noticeable on two-dimensional PCoA plots (34) and three-dimensional PCoA (5). Thus, the Hidalgo-Cantabrana et al. survey (5) is a single example where patients had a noticeable gut community shift (inferred by an unsupervised PCoA approach) and diversity decline.

Despite overall similarities in gut communities, one can still test for modest but statistically significant differences using tools such as LefSe, ANCOM, and DESeq. Thus, previous studies, including ours (Fig. 2), highlighted modest differences in abundance for different bacterial taxa (Table S1). While individual species and genera were not reproducible across studies, one unifying feature was an increase in *Firmicutes* (*Bacillota*) and a decrease in *Bacteroides* (*Bacteroidota*). However, most published studies tried to interpret "psoriasis-associated" taxa (species, genera, and families) in light of previously published associations with other inflammatory diseases. Based on such circular evidence, it is challenging to delineate normal commensals from pathobionts. Therefore, to firmly conclude gut dysbiosis, it is still difficult to address the essential criteria—evidence for depleted commensals and pathobiont bloom, not to mention quantifying bacterial mass.

Our study identified seven bacterial taxa that demonstrated statistically significant enrichment in patients (Fig. 2). Still, our analyses showed that half of them (four out of seven, i.e., *M. elsdenii*, *C. mitsuokai*, *Eubacterium* sp. *CAG:180*, and *R. mucilaginosa*) could be better explained by the elevated fecal calprotectin in patients (calprotectin > 35, Fig. 3C). Elevated calprotectin, often featured in patients, might explain why these species were identified as patient associated. In agreement with this alternative explanation, one healthy outlier (see Fig. 3C) with elevated calprotectin (calprotectin > 35) demonstrated an elevated proportion of these "psoriasis-associated" species. These species likely gained relative advantage under an altered microenvironment in the intestine under higher calprotectin release either by the colon or immune cells. This altered microenvironment is also accompanied by (i) increased lactic acid release by effector immune cells and acidity (35), (ii) elevated calprotectin that starves microbes from essential metal ions, such as copper, zinc, manganese, and iron, and (iii) increased reactive oxygen species and antibacterial peptides. In this regard, it is notable that one of the calprotectin- and psoriasis-associated bacteria, *M. elsdenii*, plays a prominent role in utilizing lactate in the context of the inflamed digestive system(36)across diverse mammals (37). Human fecal-derived *M. elsdenii* has the enzymatic capacity (specialized acrylate pathway) to

utilize lactate. This human isolate is well-equipped to resist reactive oxygen species and other factors related to inflamed intestines (38). We, therefore, hypothesize that an increase in *M. elsdenii* proportion in our set of psoriasis patients might have resulted from a response to increased lactate availability due to higher release by intestinal or immune cells. Indeed, *M. elsdenii* manifests in diverse host organisms whenever mucosal tissues, such as the vagina in humans (39), or rumen in dairy cattle, are inflamed and acidified by lactate (37). The functional impact on the host is not clear. On the one hand, *M. elsdenii* is considered a probiotic that alleviates lactic acidosis in the digestive tract (40) of animals. On the other hand, *M. elsdenii* can cause dendritic cell maturation and increased pro-inflammatory cytokines. The main question, however, remains unanswered. What comes first? Is it detrimental behavior of patient-associated bacteria that causes a higher immune cell effector activity and possible low-grade inflammation in the gut (and possible distal effects in the skin)? Or is it systemic inflammation in psoriasis patients that causes low-grade inflammation in the intestine and gut microbiome response?

## Limitations of the study and outlook

Thus, accumulating observational studies on gut microbiomes can be interpreted in both ways: rare commensals responding to the altered inflamed environment or pathobionts that cause inflammation in the gut. These two alternatives, however, cannot be distinguished in case-control studies like ours and those summarized in Table S1. We, therefore, caution that it is premature to jump to conclusions such as "dysbiotic gut in psoriasis" and its "role in psoriasis" based on published taxonomic alterations and correlations. Alternative explanations are equally possible since murine skin inflammation models suggest that the gut-skin axis is a bidirectional communication that influences both sides. For example, it was shown that skin inflammation induced by imiquimod (IMQ) leads to alterations in the gut microbiome in the animal model of psoriasis-like inflammation (41, 42). Conversely, when probiotics modulate the gut microbiome in murine models (IMQ induced), gut community alterations lead to dampened skin inflammation (43).

While the causal role of the gut microbiome remains unclear, we now know that some psoriasis patients tend to benefit from adjuvant therapy with oral probiotics. For instance, a recent meta-analysis showed that some psoriasis patients clear more effectively when treated with oral probiotics. Namely, when probiotics are used as an adjuvant therapy, patients, on average, demonstrate decreased PASI compared with the placebo group (44). Importantly, not all patients demonstrate improvement, but somewhat higher efficacy is evident from pooled data. This is consistent with our findings and reviewed studies (Table S1). Namely, patients with a strongly disturbed gut microbiome that would be visible in the large intestine are rare. Instead, psoriasis patients are heterogeneous and show varying signs of low-grade inflammation and gut community shifts, presumably driven by inflammation. Our findings suggest that for probiotics to be successful in adjuvant therapy, patients must be pre-selected based on microbiome-based evidence for gut alterations. To further increase efficacy, one must understand whether bacterial species enriched in patients can resist some of the effector activities of the administered probiotics, such as lactic acid-producing *Lactobacillus* species. For instance, *M. elsdenii,* identified in our study, is known for its resistance to high lactic acid concentrations.

Like many other studies in the field, our findings are based on fecal-derived data. It is increasingly recognized that fecal-derived microbiome is dominated by the large intestinal community and, to a lesser extent, by small intestinal species (45). In fecal material, a larger number and higher mass of colon microbes ($10^{10}$–$10^{11}$ CFU/mL) dominate over 1,000 times smaller ($10^{11}/10^8 = 1,000$) mass of the small intestinal community ($10^{4-5}$ CFU/mL in the duodenum to $10^{7-8}$ CFU/mL in the distal ileum). In addition, modest community shifts in the small intestine would be missed when applying corrections for multiple tests. Multiple tests increase due to hundreds of colon microbes in the combined fecal material. Recent studies based on ingestible pills

showed that fecal material provides limited resolution to understanding the natural variability of the human intestine's bacterial, viral, proteomic, and metabolic content. Host-microbe interactions in the small intestine are believed to be more relevant to host immunity disturbances. Therefore, non-invasive methods to sample the small intestine hold promise in overcoming the current limitations of fecal-based microbiome studies (46, 47). Thus, the lack of clear and consistent evidence for large intestinal dysbiosis and inflammation (calprotectin > 60–100) in psoriasis does not mean that other segments of the intestine do not experience inflammation. Murine models (imiquimod induced) of skin inflammation demonstrated that skin inflammation leads to pronounced changes in the small intestine but not in the large intestine. Specifically, in these animals, the small intestine but not the large intestine demonstrated elevated calprotectin, abnormal villous architecture, reduced intestinal barrier integrity, and pronounced gut community alteration (48). One can hypothesize that seemingly less pronounced alterations of the fecal-based gut microbiome in psoriasis can be partly explained by the nuances of the gut-skin axis. Indeed, inference about the large intestine based on fecal material is at odds with direct evidence of small intestinal inflammation in psoriasis patients. Namely, endoscopic examination of the ileum (lower portion of the small intestine) in psoriasis patients demonstrated clear signs of increased small intestinal inflammation. This study revealed that with increasing severity of psoriasis, there are growing signs of inflammation, such as shorter crypts, degraded villi, and higher immune cell infiltration (49). In that study, patients with mild psoriasis had irritable bowel syndrome. In contrast, patients with moderate psoriasis had non-ulcerative colitis and terminal ileitis, and patients with severe psoriasis demonstrated non-ulcerative colitis (49). Thus, findings from endoscopic examination in patients and murine models likely explain why we observe markers of low-grade intestinal inflammation in some patients but only moderate signals of microbiome shift in the large intestine, inferred from the fecal microbiome. In summary, the fecal-based microbiome profile is likely a poor proxy for hard-to-access segments of the small intestine, where inflammatory processes might be running. Novel non-invasive approaches, such as ingestible pills, can provide novel insight into the possible role of gut microbiome in psoriasis (50). We conclude that our quest to resolve the question of potential correlation or even connection between the gut microbiome and psoriasis is only at the beginning.

## ACKNOWLEDGMENTS

We thank Darya Terentyeva from Saint Petersburg Pasteur Institute for her assistance with metagenomic data processing.

This research was supported by the Russian Science Foundation, grant number 24-24-00087 (clinical assessment and recruitment of patients, collection of biological samples, DNA extraction, and shotgun sequencing). B.Y. and L.D. were supported by Saint-Petersburg State University research project 124032000041-1. B.Y. and M.Y. acknowledge support from Saint Petersburg State University grant ID PURE 95444211. V.B. was supported by the Ministry of Science and Higher Education of the Russian Federation projects FSMG-2023-018 (agreement #075-03-2023-106/12) and (agreement #FGFG-2024-002).

The funders had no role in study design, data collection, interpretation, or the decision to submit the work for publication.

B.Y. and M.Y. conceptualized the study, acquired funding, administrated the project, and supervised the study. B.Y., L.D., R.A., G.S., A.B., and V.B. performed formal analysis. B.Y., A.B., N.N., L.D., R.A., M.Y., and V.B. performed the investigation. A.B., N.N., G.S., R.A., F.B., and M.Y. curated the data. A.B., L.D., G.S., and V.B. provided software. N.N., R.A., F.B., and M.Y. provided resources. B.Y., A.B., N.N., L.D., and M.Y. wrote the original draft. B.Y., F.B., M.Y., and V.B. reviewed and edited the manuscript.

## AUTHOR AFFILIATIONS

[1]Institute of Translational Biomedicine, Saint-Petersburg State University, Saint-Petersburg, Russia

[2]Department of Genetics and Biotechnology, Saint-Petersburg State University, Saint-Petersburg, Russia

[3]SCAMT institute, ITMO University, Saint Petersburg, Russia

[4]Republican Dermatovenerologic Dispensary, Ufa, Russia

[5]Multiomics Laboratory, Moscow Institute of Physics and Technology, Moscow, Russia

[6]Medical Genetics Laboratory, Federal Research Center for Innovator and Emerging Biomedical and Pharmaceutical Technologies, Moscow, Russia

[7]Institute of Higher Nervous Activity and Neurophysiology of RAS, Moscow, Russia

[8]Department of Laboratory Medicine, Bashkir State Medical University, Ufa, Russia

[9]Republic Medical Genetic Centre, Ufa, Russia

## AUTHOR ORCIDs

Bayazit Yunusbayev  http://orcid.org/0000-0002-6035-8763

## FUNDING

| Funder | Grant(s) | Author(s) |
|---|---|---|
| Russian Science Foundation (RSF) | 24-24-00087 | Bayazit Yunusbayev |
| | | Milyausha Yunusbaeva |
| Saint Petersburg State University (SPbU) | project 124032000041-1 | Bayazit Yunusbayev |
| | | Lavrentii Danilov |
| Saint Petersburg State University (SPbU) | PURE ID 95444211 | Bayazit Yunusbayev |
| | | Milyausha Yunusbaeva |
| Ministry of Science and Higher Education of the Russian Federation (Minobrnauki of Russia) | projects FSMG-2023-018 (agreement # 075-03-2023-106/12) and (agreement # FGFG-2024-002) | Viktor Bogdanov |

## AUTHOR CONTRIBUTIONS

Bayazit Yunusbayev, Conceptualization, Formal analysis, Funding acquisition, Investigation, Project administration, Supervision, Writing – original draft, Writing – review and editing | Anna Bogdanova, Data curation, Formal analysis, Investigation, Software, Visualization, Writing – original draft | Nadezhda Nadyrchenko, Data curation, Investigation, Resources, Writing – original draft | Lavrentii Danilov, Formal analysis, Investigation, Software, Writing – original draft | Viktor Bogdanov, Formal analysis, Investigation, Software, Writing – review and editing | Grigory Sergeev, Data curation, Formal analysis, Software | Radick Altinbaev, Data curation, Formal analysis, Investigation, Resources | Fanil Bilalov, Data curation, Resources, Writing – review and editing | Milyausha Yunusbaeva, Conceptualization, Data curation, Funding acquisition, Investigation, Project administration, Resources, Supervision, Writing – original draft, Writing – review and editing

## DATA AVAILABILITY

The shotgun metagenomic data used in this study are available in the NCBI database under BioProject accession codes PRJNA1102742 and PRJNA1061168. BioProject PRJNA1061168 stores shotgun metagenomic data for 47 healthy donors collected and processed uniformly by our group but published as part of another case-control study (22). Both data sets are available for research purposes upon request. Requests should be directed to the corresponding author, Bayazit Yunusbayev (yunusbb@gmail.com).

Bioinformatic analyses and code used and implemented in this study are available at GitHub: https://github.com/Annanielle/Gut-microbiota-in-Psoriasis and https://github.com/yunusbb/Gut-microbiota-in-Psoriasis.

## ETHICS APPROVAL

Study approval was obtained from the Ethics Committee of the Republican Dermatovenerologic Dispensary 16.07.2018 no. 07/20 (Ufa, Russia). The study was conducted in accordance with the Declaration of Helsinki. All study participants completed and signed an informed consent to participate in this research project and publish the results. All clinical data were depersonalized.

## ADDITIONAL FILES

The following material is available online.

### Supplemental Material

**Supplemental figures (Spectrum01382-24-s0001.pdf).** Fig. S1 to S5.
**Table S1 (Spectrum01382-24-s0002.docx).** Short summaries of 12 published case-control gut microbiome studies.
**Table S2 (Spectrum01382-24-s0003.docx).** Top 14 bacterial species with abundance shifts in patients based on matched-pair Wilcoxon test.
**Table S3 (Spectrum01382-24-s0004.docx).** Top bacterial species associated with a linear increase in calprotectin.

### Open Peer Review

**PEER REVIEW HISTORY (review-history.pdf).** An accounting of the reviewer comments and feedback.

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
