## [Reviewer comments · Microbiology Spectrum]

Microbiology Spectrum

Gut dysbiosis narrative in psoriasis: matched-pair approach identifies only subtle shifts correlated with elevated fecal calprotectin

Bayazit Yunusbayev, Anna Bogdanova, Nadezhda Nadyrchenko, Lavrentii Danilov, Viktor Bogdanov, Grigorii Sergeev, Radick Altinbaev, Fanil Bilalov, and Milyausha Yunusbaeva

Corresponding Author(s): Bayazit Yunusbayev, Saint-Petersburg State University

Review Timeline:

Submission Date:	June 7, 2024
Editorial Decision:	July 22, 2024
Revision Received:	October 1, 2024
Editorial Decision:	October 13, 2024
Revision Received:	November 2, 2024
Accepted:	November 6, 2024

Editor: Jennifer Auchtung

Reviewer(s): Disclosure of reviewer identity is with reference to reviewer comments included in decision letter(s). The following individuals involved in review of your submission have agreed to reveal their identity: Jian Zhou (Reviewer #1)

Transaction Report:

DOI: <https://doi.org/10.1128/spectrum.01382-24>

Re: Spectrum01382-24 (Gut dysbiosis narrative in psoriasis: matched-pair approach identifies only subtle shifts correlated with elevated fecal calprotectin)

Dear Dr. Bayazit Yunusbayev:

Thank you for the privilege of reviewing your work. Below you will find my comments, instructions from the Spectrum editorial office, and the reviewer comments.

Please thoroughly address the reviewers' comments. Specifically, both reviewers suggest potential limitations of the study that should be clearly indicated in your discussion. Also, please ensure that data in NCBI will be publicly available before the manuscript is published, as this is a requirement for Microbiology Spectrum.

Revision Guidelines

Sincerely,
Jennifer Auchtung
Editor
Microbiology Spectrum

Reviewer #1 (Comments for the Author):

Overall Review

The paper titled "Gut dysbiosis narrative in psoriasis: matched-pair approach identifies only subtle shifts correlated with elevated

fecal calprotectin" investigates the gut microbiome alterations in psoriasis patients. By employing a case-control study with deep sequencing of fecal samples from 53 psoriasis patients and 47 healthy donors, the authors reconstruct the strain/species-level content of the gut microbiome. The study uses a matched-pair approach to adjust for microbiome variability and enhance statistical power, ultimately identifying subtle increases in specific bacterial species associated with intestinal and systemic inflammation. The findings suggest that gut microbiome alterations in psoriasis are more likely responses to inflammation rather than direct causes of dysbiosis.

Paper Strengths

Innovative Methodology: The study employs a matched-pair approach, which is innovative in adjusting for microbiome variability and increasing the power to detect subtle taxonomic shifts.

Comprehensive Data Analysis: Deep sequencing and rigorous data preprocessing, including quality control and decontamination, provide a high-resolution view of the gut microbiome.

Insightful Empirical Findings: The study identifies specific bacterial species correlated with elevated biomarkers of inflammation, offering new insights into the gut microbiome's role in psoriasis.

Well-Structured Literature Review: The authors provide a thorough review of previous studies, highlighting inconsistencies and gaps in the current understanding of gut dysbiosis in psoriasis.

Clear and Detailed Methodology: The paper includes detailed descriptions of the sample collection, sequencing, and data analysis methods, facilitating reproducibility.

Paper Weaknesses

Limited Sample Size: Although the study uses a matched-pair approach to enhance power, the total sample size of 100 donors may still limit the generalizability of the findings.

Focus on Large Intestine: The study's focus on fecal samples may miss critical microbiome alterations in the small intestine, which could be relevant for psoriasis.

Lack of Longitudinal Data: The cross-sectional nature of the study does not allow for assessment of temporal changes in the gut microbiome and their potential causal role in psoriasis.

Incomplete Discussion of Functional Implications: While the study identifies taxonomic shifts, it provides limited discussion on the functional implications of these shifts for psoriasis pathogenesis.

Potential Confounding Factors: Differences in age and BMI between patients and healthy controls may confound the results, and the study's adjustments for these variables could be more robust.

Questions to Authors and Suggestions for Rebuttal

Can you provide more details on how the matched-pair approach was implemented and validated?

How do you address potential confounding factors such as age and BMI differences between patients and healthy controls?

Could you elaborate on the functional implications of the identified bacterial species, particularly *Megasphaera elsdenii* and *Eubacterium* CAG 180, in the context of psoriasis?

Have you considered longitudinal studies to assess temporal changes in the gut microbiome in psoriasis patients?

What are the potential limitations of using fecal samples to infer microbiome alterations in the small intestine, and how might future studies address this?

Overall Score

Score: 7

Reasoning: The paper provides valuable insights into the gut microbiome's role in psoriasis using innovative methods and comprehensive data analysis. However, limitations such as the sample size, focus on the large intestine, and lack of longitudinal data warrant further investigation to fully understand the findings' implications

Reviewer #2 (Comments for the Author):

This study examines gut microbiome alterations in psoriasis, analyzing fecal samples from 53 psoriasis patients and 47 healthy donors. Using a matched-pair approach, the research found no significant differences in overall microbiome diversity or

structure. However, specific bacteria like *Megasphaera elsdenii* and *Eubacterium* CAG 180 were more abundant in patients and linked to inflammation markers. These findings suggest that microbiome changes in psoriasis are likely a response to inflammation rather than its cause, challenging the notion of significant dysbiosis in the disease.

While the analytical approach is interesting, it fails to validate the findings with independent datasets. Additionally, using the matched-pair approach could potentially force the identification of microbial features associated with inflammation, warranting cautious interpretation of the results

Specific comments

The shotgun metagenomic dataset PRJNA1102742 should be made available to the research community if the paper is accepted for publication.

The ethics statement must report all relevant information, including approval numbers and the ethics committee that approved the study

The authors state that there is a lack of consistency and clear demonstration that gut dysbiosis is associated with psoriasis in both human and mouse studies. However, this study remains observational and does not provide any mechanistic demonstration that the gut microbiome may or may not be involved in psoriasis pathogenesis. I suggest reconsidering the conclusions of the study accordingly.

The importance section never refers to psoriasis. Moreover, on what metrics did the authors decide that the sample size of this study is sufficient? $n=53$ patients is not a large enough sample to make any generalization, especially considering that this is a monocentric study. This is significant given that gut microbiome composition is also affected by geography. Did the authors try to validate their findings using independent cohorts? Are there any deposited metagenomics datasets from psoriasis patients that can be used for this purpose?

Introduction: The authors should expand their introduction according to the studies cited in Table S1 and elaborate specifically on why these studies are not consistent beyond the sample size, which is not significantly smaller than the sample size of this study.

It is important to report the most recent studies on this topic, such as PMID: 38441468.

Line 121: Which estimates?

Line 176: Although the authors acknowledge that LEfSe is less conservative than other tools for inferring differences between groups, it is advisable to use LEfSe on taxa that have first been selected according to FDR-corrected pairwise comparisons to minimize the possibility of obtaining false positives. Alternatively, use LEfSe and then consider as significantly enriched only those taxa that are significant by both LEfSe and FDR-corrected pairwise comparison analysis.

Line 177: Although constructing matched pairs with similar taxonomic profiles may help in discovering hidden biology due to high inter-sample variability among donors irrespective of their health status, how does using this approach allow for general assumptions on alterations of the gut microbiome in association with psoriasis? I do not understand why a healthy subject with a similar microbiome to a psoriasis patient should inform on possible taxa that are then specifically enriched or depleted in patients. What variables could make the microbiomes of matched pairs similar?

Line 279: Although psoriasis patients have significantly higher levels of fecal calprotectin (which test was used? Paired between matched-control pairs or unpaired?), only a small fraction of them have $FC > 50 \mu\text{g/g}$ and only four patients $> 100 \mu\text{g/g}$. This is not enough to infer that these patients are characterized by an active status of intestinal inflammation that can be associated with some bugs without any experimental demonstration of the proinflammatory role of the gut microbes identified in this study.

Comments and Suggestions for the Author:

Overall Review

The paper titled "Gut dysbiosis narrative in psoriasis: matched-pair approach identifies only subtle shifts correlated with elevated fecal calprotectin" investigates the gut microbiome alterations in psoriasis patients. By employing a case-control study with deep sequencing of fecal samples from 53 psoriasis patients and 47 healthy donors, the authors reconstruct the strain/species-level content of the gut microbiome. The study uses a matched-pair approach to adjust for microbiome variability and enhance statistical power, ultimately identifying subtle increases in specific bacterial species associated with intestinal and systemic inflammation. The findings suggest that gut microbiome alterations in psoriasis are more likely responses to inflammation rather than direct causes of dysbiosis.

Paper Strengths

1. **Innovative Methodology**: The study employs a matched-pair approach, which is innovative in adjusting for microbiome variability and increasing the power to detect subtle taxonomic shifts.
2. **Comprehensive Data Analysis**: Deep sequencing and rigorous data preprocessing, including quality control and decontamination, provide a high-resolution view of the gut

microbiome.

3. **Insightful Empirical Findings**: The study identifies specific bacterial species correlated with elevated biomarkers of inflammation, offering new insights into the gut microbiome's role in psoriasis.

4. **Well-Structured Literature Review**: The authors provide a thorough review of previous studies, highlighting inconsistencies and gaps in the current understanding of gut dysbiosis in psoriasis.

5. **Clear and Detailed Methodology**: The paper includes detailed descriptions of the sample collection, sequencing, and data analysis methods, facilitating reproducibility.

Paper Weaknesses

1. **Limited Sample Size**: Although the study uses a matched-pair approach to enhance power, the total sample size of 100 donors may still limit the generalizability of the findings.

2. **Focus on Large Intestine**: The study's focus on fecal samples may miss critical microbiome alterations in the small intestine, which could be relevant for psoriasis.

3. **Lack of Longitudinal Data**: The cross-sectional nature of the study does not allow for assessment of temporal changes in the gut microbiome and their potential causal role in psoriasis.

4. **Incomplete Discussion of Functional Implications**: While the study identifies taxonomic shifts, it provides limited discussion on the functional implications of these shifts for psoriasis pathogenesis.

5. **Potential Confounding Factors**: Differences in age and BMI between patients and healthy controls may confound the results, and the study's adjustments for these variables could be more robust.

Questions to Authors and Suggestions for Rebuttal

1. Can you provide more details on how the matched-pair approach was implemented and validated?

2. How do you address potential confounding factors such as age and BMI differences between patients and healthy controls?

3. Could you elaborate on the functional implications of the identified bacterial species, particularly *Megasphaera elsdenii* and *Eubacterium CAG 180*, in the context of psoriasis?

4. Have you considered longitudinal studies to assess temporal changes in the gut microbiome in psoriasis patients?

5. What are the potential limitations of using fecal samples to infer microbiome alterations in the small intestine, and how might future studies address this?

Overall Score

Score: 7

****Reasoning****: The paper provides valuable insights into the gut microbiome's role in psoriasis using innovative methods and comprehensive data analysis. However, limitations such as the sample size, focus on the large intestine, and lack of longitudinal data warrant further investigation to fully understand the findings' implications.

Confidential remarks for the Editors:

The paper "Gut dysbiosis narrative in psoriasis: matched-pair approach identifies only subtle shifts correlated with elevated fecal calprotectin" presents a sophisticated and

detailed analysis of the gut microbiome in psoriasis patients. The authors use an innovative matched-pair approach to control for microbiome variability, enhancing the statistical power to detect subtle shifts in bacterial populations correlated with inflammatory biomarkers.

Strengths:

Methodological Rigor: The use of deep sequencing and rigorous data preprocessing ensures high-quality data, providing a robust foundation for the study's conclusions.

Insightful Findings: The identification of specific bacterial species correlated with elevated inflammation biomarkers adds valuable insights to the ongoing discourse on the gut-skin axis in psoriasis.

Comprehensive Review and Analysis: The paper provides a thorough review of existing literature, highlighting inconsistencies and addressing gaps with its findings.

Weaknesses:

Sample Size and Generalizability: The relatively small sample size (100 donors) may limit the generalizability of the findings. Larger, multicenter studies would be beneficial for validating these results.

Cross-sectional Design: The lack of longitudinal data limits the ability to assess temporal changes and potential causal relationships in the gut microbiome and psoriasis.

Functional Analysis: The paper could benefit from a more detailed discussion of the

functional implications of the identified taxonomic shifts for psoriasis pathogenesis.

Recommendation:

Given the paper's innovative approach and significant contributions to understanding the gut microbiome's role in psoriasis, I recommend it for publication, with the suggestion that the authors address the mentioned weaknesses, particularly regarding the discussion on functional implications and potential confounding factors. Additionally, encouraging the authors to consider longitudinal studies in future research could provide more comprehensive insights into the dynamic nature of gut microbiome changes in psoriasis.

Overall, this paper represents a meaningful addition to the field and is likely to stimulate further research and discussion.

We thank the reviewers for carefully reading the manuscript and for their constructive remarks, which helped us improve it.

Please find below our point-by-point responses to all the comments concerning the original submission.

Note that the line numbers in our responses refer to the manuscript's **CLEAN VERSION (not Marked-Up Manuscript)**.

A point-by-point response to Reviewer #1

1. Can you provide more details on how the matched-pair approach was implemented and validated?	Thank you for your question. Since the matched-pair approach was introduced earlier for microbiome studies and represents a classical approach, we haven't considered validation. That said, we would also be curious to see dedicated studies that use mock communities to test matched-pair approach limits with complex communities. We now rephrased the text to emphasize that we are not introducing a novel approach but using an earlier established approach. Lines: 280-282. We have now provided all our in-house scripts and notebooks that can be used to reproduce the matched-pair approach on our GitHub page (https://github.com/Annanielle/Gut-microbiota-in-Psoriasis/tree/main). Specifically, you can find the detailed implementation of the matched-pair analysis in the notebook "3_matching_pairs_analysis.ipynb" We added a reference to our GitHub page in the Materials and Methods section.
2. How do you address potential confounding factors such as age and BMI differences between patients and healthy controls?	Indeed, mean values of age and BMI slightly differ (statistically significant) between cases and controls (Table 1). In response to the reviewer's concern, we added tests to see whether age and BMI correlate with microbial variation in our

	dataset. We computed principal components of microbial variation, projected donors on computed PCs, and, for each donor, superimposed age, BMI, and sex (Figure S4, panels A, B, and C, respectively). Neither age, BMI, nor sex showed a correlation with principal components of microbiome variation (Figure S4). We added these results to our main text: Lines: 248-251
3. Could you elaborate on the functional implications of the identified bacterial species, particularly Megasphaera elsdenii and Eubacterium CAG 180, in the context of psoriasis?	In the absence of functional studies on human-derived Megasphaera elsdenii, we draw on human and animal-based studies whenever this species has been isolated and well-studied. From animal studies, we know that M. elsdenii thrives in the inflamed gut when animals suffer from lactate accumulation and acidosis. This role seems universal for diverse mammals(1). This specific role is explained by adaptations to efficiently utilize lactate via the acrylate pathway in an acidic environment(2). Because of this capacity, M. elsdenii is proposed as a probiotic that can alleviate animal lactate acidosis (3). It was shown that human gut-derived M. elsdenii also has this genetic pathway to utilize lactate and harbor adaptations to withstand oxidative stress, bile acid, and other gut-associated factors (4). We, therefore, hypothesize that M. elsdenii might increase in psoriasis patients as a response to lactate accumulation in the patient's intestine. We have now added our interpretation in the Discussion section. Lines: 404-417.
4. Have you considered longitudinal studies to assess temporal changes in the gut microbiome in psoriasis patients?	We agree that obtaining multiple samples over time is desirable when studying gut microbiome. However, in clinical practice, patients only have one to two days before they start taking medications (retinoids, immunosuppressive drugs, cytostatics, etc.) when they arrive in the hospital for

	treatment. Therefore, due to treatment regimen logistics, fecal sample collection was performed on the first two days of patient admission.
5. What are the potential limitations of using fecal samples to infer microbiome alterations in the small intestine, and how might future studies address this?	We now expanded our discussion by adding more details about the limitations of using fecal samples. We believe novel, non-invasive, ingestible pills can address the long-standing issue of accessing small intestinal mucosal material. Please see the Discussion section. Lines: 432-456.

We thank the reviewers for carefully reading the manuscript and for their constructive remarks, which helped us improve it.

Please find below our point-by-point responses to all the comments concerning the original submission.

Note that the line numbers in our responses refer to the manuscript's **clean version (not Marked-Up Manuscript)**.

A point-by-point response to Reviewer #2

1. The ethics statement must report all relevant information, including approval numbers and the ethics committee that approved the study	We added approval numbers in the sections "Ethics statement" Lines 37-41
2. The authors state that there is a lack of consistency and clear demonstration that gut dysbiosis is associated with psoriasis in both human and mouse studies. However, this study remains observational and does not provide	We agree that by highlighting inconsistencies in previous studies, we haven't provided mechanistic evidence to test the alternative view. We revised the main text and ensured that our statements were balanced

any mechanistic demonstration that the gut microbiome may or may not be involved in psoriasis pathogenesis. I suggest reconsidering the conclusions of the study accordingly.	whenever we discussed the potential role of the gut microbiome in psoriasis. For instance, we noticed that our conclusions in the Abstract are fairly strong in favoring the alternative view. We have rephrased conclusions to sound more balanced: Specifically, Lines: 63-65 Lines: 424-431
3. The importance section never refers to psoriasis. Moreover, on what metrics did the authors decide that the sample size of this study is sufficient? n=53 patients is not a large enough sample to make any generalization, especially considering that this is a monocentric study. This is significant given that gut microbiome composition is also affected by geography. Did the authors try to validate their findings using independent cohorts? Are there any deposited metagenomics datasets from psoriasis patients that can be used for this purpose?	Many thanks for bringing this to our attention. We have rephrased the importance section to refer to gut microbiome in psoriasis patients. Lines: 74-81. We have now performed effect size calculations for psoriasis and power estimates using a novel computational tool for microbiome data(5). Our estimates suggest that psoriasis has a small effect size on gut microbiome structure (Cohen's $d=0.37$), which is comparable to other autoimmune diseases (Cohen's $d=0.20$) (5). As a rule of thumb, Cohen's d values of 0.2, 0.5, and 0.8 are generally considered "small," "medium", and "large" effect sizes, respectively. Thus, only moderate alterations are expected when psoriasis is considered a potential factor influencing the microbiome. Hence, there are only a few group-wise alterations/shifts from healthy controls. Indeed, most studies on psoriasis, summarised in Table S1, did not find clear group-wise separation. Some studies reported visible shifts from healthy controls. Given the small effect size, detecting group-wise shift with power $P=0.80$ would require a sample size larger than $N=200$ (See Power curve plot in Figure S1). While our sample size of $N=100$ is insufficient to detect such small shifts using conventional group-wise tests,

	we adopted a more productive approach. We increased statistical power to detect small systematic shifts in patients by focusing on differences with their matched pairs among healthy controls. We agree that validation with an independent cohort would be desirable; however, with the budget of our small pilot study, we could not afford to sample another cohort. Also, only two out of many published datasets are truly accessible but are not comparable in sequencing depth. We have added a description of power estimates in the Materials and Methods section and our rationale for applying the matched-pair approach. Lines: 131-141
4. Introduction: The authors should expand their introduction according to the studies cited in Table S1 and elaborate specifically on why these studies are not consistent beyond the sample size, which is not significantly smaller than the sample size of this study.	In response to the reviewer's comment, we have revised and expanded the Introduction section based on previous studies (summarized in Table S1). Still, we tried to keep our focus on the two major issues pertinent to our study - evidence for dysbiosis and lack of reproducible disease-associated species. Lines: 87-94 and 103-109
5. It is important to report the most recent studies on this topic, such as PMID: 38441468.	Thank you for pointing to this new work. We have added the new study to our structured review in Table S1.
6. Line 121: Which estimates?	We have now updated our statistical power estimates using the evident software (https://github.com/biocore/evident), a computational tool designed for microbiome data (5).

	Please see our revised text on the new power analysis (Figure S1) in the Materials and Methods section. Lines 131-141 We also provide a detailed notebook on our power analysis at GitHub: https://github.com/Annanielle/Gut-microbiota-in-Psoriasis/blob/main/power_analysis/power_analysis_clr_sp.ipynb.
7. Line 176: Although the authors acknowledge that LEfSe is less conservative than other tools for inferring differences between groups, it is advisable to use LEfSe on taxa that have first been selected according to FDR-corrected pairwise comparisons to minimize the possibility of obtaining false positives. Alternatively, use LEfSe and then consider as significantly enriched only those taxa that are significant by both LEfSe and FDR-corrected pairwise comparison analysis.	Many thanks for highlighting this uncertainty in the data presentation. We agree that prefiltering with FDR is a logical step to minimize false positives, but it also incurs a subtle issue for biological inference. Namely, LEfSe uses linear combinations of species, and prefiltering can remove some species that contribute to biologically plausible linear combinations. We would also like to highlight that in the main text and discussion, our major findings are based on the Matched-pair Wilcoxon test (Figure 2). However, given that most earlier studies used LEfSe, we felt that LEfSe might help allow for comparison. So, we deliberately placed LEfSe results in the supplementary. Reference in the text: Lines: 288-289.
8. Line 177: Although constructing matched pairs with similar taxonomic profiles may help in	We agree that the procedure and the name of the approach, "matched-pair" sounds counter-intuitive. However, the

discovering hidden biology due to high inter-sample variability among donors irrespective of their health status, how does using this approach allow for general assumptions on alterations of the gut microbiome in association with psoriasis? I do not understand why a healthy subject with a similar microbiome to a psoriasis patient should inform on possible taxa that are then specifically enriched or depleted in patients. What variables could make the microbiomes of matched pairs similar?	method is classical and has been established in biology and medicine. To give the intuition, it would be helpful to clarify some nuances. When we seek matched pairs for patients, we never find healthy donors with perfectly matching microbiomes. In fact, for each pair with "healthy subject with a similar microbiome," there are always differences in patients. When compared across pairs, small differences can be spotted if they are systematic, but normally, they are buried among dominant and highly variable species. So, one needs to look at multiple matched pairs (each different from one another) to find systematic shifts in patients that can add up to a significant Wilcoxon test. We have now added some introductory sentences to the Materials and Methods section. Lines: 200-202
9. Line 279: Although psoriasis patients have significantly higher levels of fecal calprotectin (which test was used? Paired between matched-control pairs or unpaired?), only a small fraction of them have FC > 50 µg/g and only four patients > 100 µg/g. This is not enough to infer that these patients are characterized by an active status of intestinal inflammation that can be associated with some bugs without any experimental demonstration of the proinflammatory role of the gut microbes identified in this study.	Statistical analysis was done with unpaired Wilcoxon test (wilcox.test R function). We now added clarification that a higher subset of patients with elevated calprotectin explains higher average calprotectin in patients: Lines: 307-309. We agree that elevated calprotectin does not imply "an active status of intestinal inflammation." In fact, we tried to distinguish between "elevated calprotectin" and increased calprotectin, which is used to diagnose intestinal inflammation. This was defined in the following sentences: Lines 309-312 Then, throughout the text, we deliberately use an "elevated" level of calprotectin,

	which follows this operational definition. While we do not conclude intestinal inflammation, we have reasons to hypothesize traces of "low-grade inflammation." We tried to make a distinction between inference and hypothesis by rephrasing all such sentences. Sentences: Lines: 65, 77-79, 417-420 Finally, we conclude at the end of the discussion, "Thus, the lack of clear and consistent evidence for large intestinal dysbiosis and strong inflammation (calprotectin > 60-100)..." Lines: 444-446
--	--

1. Monteiro HF, Faciola AP. 2020. Ruminal acidosis, bacterial changes, and lipopolysaccharides. *J Anim Sci* 98.
2. Louis P, Duncan SH, Sheridan PO, Walker AW, Flint HJ. 2022. Microbial lactate utilisation and the stability of the gut microbiome. *Gut Microbiome* 3:e3.
3. Susanto I, Wiryawan KG, Suharti S, Retnani Y, Zahera R, Jayanegara A. 2023. Evaluation of *Megasphaera elsdenii* supplementation on rumen fermentation, production performance, carcass traits and health of ruminants: a meta-analysis. *Anim Biosci* 36:879–890.
4. Shetty SA, Marathe NP, Lanjekar V, Ranade D, Shouche YS. 2013. Comparative genome analysis of *Megasphaera* sp. reveals niche specialization and its potential role in the human gut. *PLoS One* 8:e79353.

5. Rahman G, McDonald D, Gonzalez A, Vázquez-Baeza Y, Jiang L, Casals-Pascual C, Peddada S, Hakim D, Dilmore AH, Nowinski B, Knight R. 2022. Scalable power analysis and effect size exploration of microbiome community differences with Evident. bioRxiv.

Re: Spectrum01382-24R1 (Gut dysbiosis narrative in psoriasis: matched-pair approach identifies only subtle shifts correlated with elevated fecal calprotectin)

Dear Dr. Bayazit Yunusbayev:

Thank you for the privilege of reviewing your work. As you will see below, you are very close to acceptance, but there are a few comments to address from reviewer 1.

Revision Guidelines

Sincerely,
Jennifer Auchtung
Editor
Microbiology Spectrum

Reviewer #1 (Comments for the Author):

Overall, this verison, is a well-executed study that addresses a timely and relevant question in the field of psoriasis research. the authors addressed all my concerns, but there still some points should be noted :

Minor Suggestions for Improvement:
Clarification in the Discussion:

While the manuscript's discussion is generally clear, I recommend slightly expanding on the potential implications of the findings. For instance, the authors could explore how their results compare with existing studies that have found stronger signals of dysbiosis in other autoimmune conditions. Additionally, it would be helpful to emphasize the clinical relevance of the subtle microbial shifts identified, particularly in the context of developing future therapeutic strategies targeting the gut-skin axis.

Figure Clarity:

The figures are generally well-constructed, but a visual summary of the matched-pair approach (e.g., a flowchart) would help readers unfamiliar with this technique better understand the methodology. This would also highlight the innovative nature of the study's design.

Language and Readability:

The manuscript is well-written, though a few complex sentences could be simplified to enhance readability, especially for non-native English speakers. For example, breaking long sentences into more concise statements would improve the overall flow of the text.

Conclusion:

In conclusion, this manuscript presents a rigorous, well-executed study with novel insights into the role of the gut microbiome in psoriasis. The use of a matched-pair approach strengthens the findings and sets this study apart from others in the field. The results are relevant and timely, particularly given the growing interest in the gut-skin axis and microbiome-related therapies in autoimmune diseases.

We thank the reviewer for taking the time to evaluate our manuscript and for collaborating to improve it.

Please find below our point-by-point responses to all the comments concerning the revised submission.

Note that the line numbers in our responses refer to the manuscript's **CLEAN VERSION (not Marked-Up Manuscript)**.

A point-by-point response to Reviewer #1

1. While the manuscript's discussion is generally clear, I recommend slightly expanding on the potential implications of the findings. For instance, the authors could explore how their results compare with existing studies that have found stronger signals of dysbiosis in other autoimmune conditions. Additionally, it would be helpful to emphasize the clinical relevance of the subtle microbial shifts identified, particularly in the context of developing future therapeutic strategies targeting the gut-skin axis.	Indeed, some autoimmune conditions have "stronger signals of dysbiosis" than psoriasis. However, as mentioned in our previous response, effect size estimates for autoimmune conditions in the American Gut project[1, 2] are comparable with those for psoriasis. From these standardized measures, it is difficult to claim that psoriasis stands out. This is puzzling and can be partly explained if we admit that microbiome studies often overstate the significance of identified microbial alterations. Nevertheless, animal models seem to explain why fecal-based microbiome studies in psoriasis did not find strong signals in the large intestine. For psoriasis, animal skin inflammation models suggest that the gut-skin axis strongly affects the small intestine but not the large intestine. As demonstrated by Kim et al., when skin inflammation is induced, the large intestine remains unchanged. On the contrary, the small intestine is visibly disturbed, as evidenced by elevated calprotectin, abnormal villous architecture, reduced intestinal barrier integrity, and pronounced gut community alterations [3]. Thus, animal models suggest that the gut-skin axis is preferentially confined to the small intestine. These findings can explain why we might observe varying inflammation markers in some patients but little
---	---

	alterations in the large intestine (fecal-based microbiome). This nuanced connection between the small intestine and skin inflammation can also explain why there is endoscopic evidence for small intestinal inflammation in psoriasis patients. Still, most microbiome studies fail to show overt dysbiosis in the large intestine from fecal material. We conclude by hypothesizing that the gut microbiome could be disturbed in the small intestine, so further research is needed with novel sampling techniques. We have now revised our text to discuss these nuances of the gut-skin axis: Lines: 469-477, 481-490 We also updated our text to discuss possible implications for adjuvant therapy in psoriasis. Lines: 435-456.
2. Figure Clarity: The figures are generally well-constructed, but a visual summary of the matched-pair approach (e.g., a flowchart) would help readers unfamiliar with this technique better understand the methodology. This would also highlight the innovative nature of the study's design.	We have added a graphical diagram in Figure S5 to help readers understand the matched-pair approach. The diagram conveys the central idea behind the matched-pair approach. Please find Figure S5 in the combined file 'Supplementary_Figures.pdf'. References in the text: Lines: 202-203, 284.
3. Language and Readability: The manuscript is well-written, though a few complex sentences could be simplified to enhance readability, especially for non-native English speakers. For example, breaking long sentences into more concise statements would improve	Indeed, while the readability score (12) was fair for a technical academic text, the Hemingway editor (https://hemingwayapp.com/) identified several long, hard-to-read sentences. We have now edited some of the longest and most difficult-to-read sentences (over

the overall flow of the text.	19 sentences) to improve readability. Lines: 118-119, 196-197, 210-212, 225-228, 251-254, 280-282, 285-286, 302-304, 304-305, 313-317, 358-360, 368-371, 379-382, 405-409, 412-414, 419-422, 422-424, 462-464, 479-483.
---

1. Rahman G, McDonald D, Gonzalez A, et al (2022) Scalable power analysis and effect size exploration of microbiome community differences with Evident. bioRxiv 2022.05.19.492684
2. McDonald D, Hyde E, Debelius JW, et al (2018) American Gut: an Open Platform for Citizen Science Microbiome Research. mSystems. <https://doi.org/10.1128/mSystems.00031-18>
3. Kim HJ, Jang J, Na K, et al (2024) TLR7-dependent eosinophil degranulation links psoriatic skin inflammation to small intestinal inflammatory changes in mice. Exp Mol Med 56:1164–1177

Re: Spectrum01382-24R2 (Gut dysbiosis narrative in psoriasis: matched-pair approach identifies only subtle shifts correlated with elevated fecal calprotectin)

Dear Dr. Bayazit Yunusbayev:

Your manuscript has been accepted, and I am forwarding it to the ASM production staff for publication. Your paper will first be checked to make sure all elements meet the technical requirements. ASM staff will contact you if anything needs to be revised before copyediting and production can begin. Otherwise, you will be notified when your proofs are ready to be viewed.

Sincerely,
Jennifer Auchtung
Editor
Microbiology Spectrum